# Pediatric Kikuchi–Fujimoto Disease: Case Report and Review of Cutaneous and Histopathologic Features in Childhood

**DOI:** 10.3390/dermatopathology12010007

**Published:** 2025-02-13

**Authors:** Alberto Soto-Moreno, Francisco Vílchez-Márquez, María Narváez-Simón, Julia Castro-Martín, Francisco Manuel Ramos-Pleguezuelos, Agustín Soto-Díaz, Jesús Tercedor-Sánchez, Salvador Arias-Santiago

**Affiliations:** 1Dermatology Department, Hospital Universitario Virgen de Las Nieves, 18012 Granada, Spain; albertosotomoreno96@gmail.com (A.S.-M.); francisco.vilchez.marquez.sspa@juntadeandalucia.es (F.V.-M.); juliacastromart@gmail.com (J.C.-M.); agustin.soto.diaz@gmail.com (A.S.-D.); jesustercedor@gmail.com (J.T.-S.); 2Pathology Derpartment, Hospital Universitario Virgen de Las Nieves, 18012 Granada, Spainpatologiafmrp@gmail.com (F.M.R.-P.); 3Granada Institute for Biosanitary Research, 18012 Granada, Spain

**Keywords:** Kikuchi–Fujimoto disease, children, histiocytic necrotizing lymphadenitis, lupus erythematosus, skin

## Abstract

Kikuchi–Fujimoto disease (KFD) is a rare condition characterized by necrotizing lymphadenitis and fever, often associated with immune dysregulation. Histologically, it features necrotic foci with abundant histiocytes and plasmacytoid dendritic cells but notably lacks neutrophils and eosinophils. Recent evidence reveals a notable prevalence among pediatric patients, who may exhibit distinct features compared to adults. We reported the case of an 11-year-old girl presenting with persistent fever, cervical adenopathy, and a malar rash, leading to a diagnosis of KFD following lymph node biopsy, which revealed non-suppurative necrosis and histiocytic infiltration. Empirical treatment with antivirals and antibiotics was ineffective, but corticosteroid therapy achieved symptom remission. A literature review identified 48 relevant studies involving 386 pediatric cases, with histopathological findings consistent with classical descriptions of KFD. Cutaneous involvement was reported in 11.14% of cases, ranging from maculopapular rashes to lupus-like eruptions. Notable complications included the development of systemic lupus erythematous, Sjögren syndrome, and rare instances of hemophagocytic syndrome or central nervous system involvement. Kikuchi–Fujimoto disease should be considered in the differential diagnosis of pediatric patients presenting with fever and lymphadenopathy, taking into account a higher frequency of cutaneous manifestations in pediatric cases. A skin biopsy may be helpful in diagnosing KFD and provide valuable information regarding the potential risk of developing systemic lupus erythematosus in the future.

## 1. Introduction

Kikuchi–Fujimoto disease (KFD) is a rare condition characterized by subacute necrotizing lymphadenopathy and fever [1,2]. Its etiology remains unknown, although an immune-mediated pathogenesis has been proposed [2]. This hypothesis could explain the association between KFD and systemic lupus erythematosus (SLE), which relatively frequently results in the coexistence of both conditions in the same patient [2].

Histologically, KFD is characterized by histiocytic necrosis with abundant karyorrhectic nuclear debris and a notable absence of neutrophils and eosinophils [2]. Immunohistochemistry (IHC) commonly reveals histiocytes that are myeloperoxidase-positive (MPO+) and CD68+, a predominance of T-cells within the lesions, and CD123+ plasmacytoid dendritic cells surrounding the necrotic areas [1,2,3].

Although KFD was initially thought to predominantly affect young adults, recent increases in reports of pediatric cases suggest that this condition is not as uncommon in childhood as previously believed [4,5,6]. Additionally, some studies have suggested differences between pediatric and adult cases, such as a higher prevalence of cutaneous rash in children [4]. In this study, we present a case of pediatric KFD and provide a literature review to enhance the understanding of this condition in children.

## 2. Case Report

### 2.1. Clinical Presentation

We present the case of an 11-year-old female admitted to the Department of Pediatrics with a persistent fever lasting more than a month, reaching temperatures up to 39 °C, and accompanied by a cervical adenopathic conglomerate. A similar episode had resulted in hospitalization the previous month, which had resolved with intravenous (IV) methylprednisolone.

During her hospital stay, she developed a rash characterized by erosive lesions and crusts located in the bilateral ciliary and malar regions, as well as on the lobes of both ears (Figure 1). The patient was clinically diagnosed with Kaposi’s varicelliform eruption, and a PCR test for herpes simplex virus was performed on the erosive lesions. Additionally, a complete blood count (CBC), autoimmunity panel, lymphocyte subpopulation analysis, and serology for multiple infectious agents were requested. An excisional biopsy of the cervical adenopathy and a punch biopsy of the skin were also conducted. Before a definitive clinical diagnosis was established, intravenous linezolid (600 mg/12 h) and acyclovir (10 mg/kg/8 h) were administered empirically but without significant clinical improvement.

PCR testing for viruses in the erosive lesions was negative. The CBC revealed leucopenia (2750/µL) and a slight increase in acute phase reactants. Serology tests for cytomegalovirus (CMV), Epstein–Barr virus (EBV), human immunodeficiency virus (HIV), Hepatitis A, B, and C, Toxoplasma, Treponema, Bartonella, Brucella, and Leishmania were all negative. However, IgM antibodies against *Mycoplasma pneumoniae* were identified, along with positive antinuclear antibodies (ANAs) (speckled pattern, 1:160).

Although the skin biopsy demonstrated a nonspecific superficial perivascular lymphocytic infiltrate, the histological examination of the lymph node revealed findings consistent with non-suppurative necrotizing lymphadenitis with abundant histiocytes. Based on these results, a diagnosis of Kikuchi–Fujimoto disease (KFD) was made. IV methylprednisolone (40 mg/day for 5 days) was administered, leading to the remission of both systemic and cutaneous symptoms. The patient was subsequently discharged on oral prednisone (30 mg/day with gradual tapering).

The patient is currently receiving hydroxychloroquine (200 mg/day) and remains under follow-up care with the pediatric dermatology department. One month after discharge, she has not experienced any recurrence of adenopathy or fever.

### 2.2. Histopathologic Findings

The excisional biopsy revealed a lymph node measuring 1.3 × 1 cm, with a brownish surface and a typical nodular appearance.

Microscopically, the lymph node exhibited a distorted architecture, with residual lymphoid follicles observed in the cortex and expansion of the paracortical area (Figure 2). Well-defined paracortical foci of non-suppurative necrosis with areas of karyorrhexis were identified. These foci were surrounded by an infiltrate composed of immunoblasts, plasma cells, histiocytes (Figure 2B), abundant aggregates of CD123+ plasmacytoid dendritic cells (Figure 2C), and lymphocytes predominantly expressing T-cell markers (CD3, CD2, CD5, CD7, CD4, and CD8). Sparse and occasionally scattered TdT-positive elements were also observed. No polymorphonuclear cells, eosinophils, or well-formed granulomas were identified.

Immunohistochemical markers assessed included CD20, Pax-5, CD79a, CD3, CD23, Bcl-6, Bcl-2, CD10, TdT, CD99, MUM-1, CD138, CD68, S100, CD1a, CD30, CD15, CD4, CD8, CD2, CD5, CD7, Ki-67, and CD123. Immunohistochemical staining for CMV was negative. The in situ hybridization study for the detection of EBV RNA was also negative. PAS, Giemsa, Grocott, and Warthin–Starry stains, along with immunohistochemical staining for spirochetes, did not detect microorganisms in the examined sections.

The punch biopsy of the skin from the right eyelid area consisted of a 5 mm thick sample with a typical surface and included subcutaneous cellular tissue. Microscopically, a mild superficial chronic perivascular dermatitis was observed, without adnexal tropism, interface changes, basement membrane thickening, or interstitial mucin deposition (see Figure 3). IHC was performed, showing no CD123-positive dendritic cells. An additional punch biopsy sample was submitted fresh for immunofluorescence study, which yielded negative results. The immunofluorescence panel included IgA, IgG, IgM, and C3.

## 3. Literature Review

### 3.1. Methods

A bibliographic search was conducted in the Medline database through PubMed, using the following search strategy: “Kikuchi-Fujimoto disease” OR “Kikuchi’s disease” OR “Necrotizing histiocytic lymphadenitis” AND (pediatric OR childhood OR children). A total of 225 results were identified. After reviewing titles and abstracts, 60 studies were selected. Three studies were excluded because they were reviews, two studies were excluded due to lack of access, and seven studies were excluded because they were not relevant to the research topic upon full-text review (see Figure 4).

### 3.2. Results

A total of 48 studies were included in the final analysis (see Table 1): 11 were retrospective observational studies, and 37 were case reports involving one or two cases. The studies included 386 patients (209 males, 163 females, and 14 of unspecified gender). The overall weighted mean age was 12.67 years.

### 3.3. Cutaneous Manifestations

Skin manifestations were observed in approximately 11.14% of the sample. Generalized maculopapular rashes were reported in at least 40 patients [7,8,9,10,11,12,13,14], and other findings included plantar annular erythema, chilblain lupus-like rashes [15], and non-pruritic generalized rash, sparing the nasolabial folds [16]. Oral ulcers were reported in 13 patients [13,17]. The only case of malar rash concomitant with KFD developed lupus during follow-up [18].

### 3.4. Histopathology

Most studies reported common histologic findings from excisional lymph node biopsy specimens, including cortical/paracortical necrosis, karyorrhexis, the presence of histiocytes (CD68+), plasmacytoid dendritic cells (CD123+), T-cell activation, and the absence of neutrophils, eosinophils, and granulomas (see Table 1). Lin et al. described the three histologic phenotypes of KFD: proliferative, necrotizing, and xanthomatous [19]. Camacho et al. identified organized granulomas in a lymph node specimen [20]. Skin punch biopsy was performed in only one case, revealing histologic findings of lymphohistiocytic infiltrates, predominantly perivascular, without necrosis of the adjacent vascular wall [7].

### 3.5. Treatment

A significant number of cases were self-limiting and did not require specific treatment [6,8,10,11,12,19,20,21,22,23,24,25,26,27,28,29]. In severe cases, or those associated with hemophagocytic syndrome, the most frequently administered treatments were prednisone [9,11,12,16,18,19,25,30,31,32], and IV methylprednisolone sometimes combined with IV immunoglobulin [9,14,33,34,35,36,37,38,39]. Hydroxychloroquine was commonly used in cases associated with SLE or lupus-like rashes [12,13,15,16,18,19].

### 3.6. Complications

A total of 10 cases of SLE were documented [5,6,10,13,14,15,16,18,37], with only two occurring concomitantly with KFD [13,37]. Additionally, four patients developed Sjögren syndrome [14,15]. Among the reported complications, central nervous system involvement [14,23,39,40] and hemophagocytic syndrome [9,14,33,41] were particularly noteworthy.

**Table 1 dermatopathology-12-00007-t001:** Literature review. ANAs: antinuclear antibodies; CNS: central nervous system; CMV: cytomegalovirus; COVID-19: coronavirus disease 2019; EBV: Epstein–Barr virus; IV: intravenous; IVIG: intravenous immunoglobulin; MTX: methotrexate; NA: not available; SLE: systemic lupus erythematosus; and SS: Sjögren syndrome.

Study (Date)	N; Sex; Age.	Sample	Histopathology	Cutaneous Manifestations	Possible Trigger	Treatment	Associated Complications
Heldenberg et al. [21] 1996	1; male; 10.	Lymph node excisional biopsy	Cortical or paracortical necrosis. Karyorrhexis. Histiocytes (CD68+), plasmacytoid dendritic cells (CD123+), T-cell activation. Absence of neutrophils, eosinophils and granulomas	No	No	No specific therapy	No
Gonzálvez-Piñera et al. [8] 2000	1; female; 7			Generalized maculopapular rash	No	No specific therapy	No
Emir et al. [42] 2001	1; female; 14			No	No	No specific therapy	No
Kim et al. [9] 2003	1; female; 13			Generalized maculopapular rash	EBV	IVIG IV methylprednisolone	Hemophagocytic syndrome
Chiang et al. [22] 2004	1; male; 8			No	No	No specific therapy	No
Chuang et al. [10] 2005	64; 35 males; 29 females; mean age 16			Generalized maculopapular rash (4/64)	CMV (2/64)	No specific therapy	Developed SLE (1/64)
Lin et al. [6] 2005	23; 15 males; 8 females; mean age 12.8			NA	NA	No specific therapy	Developed SLE (1/23)
Park et al. [30] 2007	16; 8 males; 8 females; mean age 10.6			No	EBV (2) ANA (1)	Oral prednisone (6/16)	No
Zou et al. [11] 2009	36; 23 males; 13 females; mean age 10.1			Generalized maculopapular rash (5/36)	EBV (3/36) HSV (2/36) ANA (3/36)	No specific therapy	No
Gómez et al. [31] 2010.	1; female; 10			No	No	Oral prednisone	No
Lee et al. [5] 2012	9; 8 males;1 female; mean age 11.3			Generalized maculopapular rash (2/9)	No ANA (1/9)	High-dose corticosteroids, IVIG, plasmapheresis	Developed SLE (1/9)
Tchidjou et al. [43] 2004	1; male; 11			No	No	IV methylprednisolone	
Gonçalves et al. [40] 2014	1; male; 9			No	No	IV Dexamethasone	CNS involvement
Martins et al. [18] 2014	1; female; 12			malar rash	ANAs	Oral prednisone Hidroxychloroquine	Developed SLE
Ojeda et al. [32] 2015	1; female; 9			No	ANAs	Oral prednisone	No
Rossetti et al. [44] 2015	1; male; 11			No	No	Corticosterois	Upper airway obstruction
Altinel et al. [35] 2018	1; male; 12			No	Salmonella	IVIG	Auto-immune thyroiditis and papilloedema
Chowdhury et al. [39] 2019	1; female; 14			No	No	IV methylprednisolone	CNS involvement
Handa et al. [45] 2016	1; male; 15			No	No	Oral corticosteroids	No
Lelii et al. [23] 2018	2; male; 12; female; 16			No	CMV (1/2) ANA (1/2)	No specific therapy (1/2) Oral prednisone and cyclosporine (1/2)	CNS involvement in 1 patient
Singh et al. [24] 2019	1; male; 11			No	No	No specific therapy	No
Quadir et al. [25] 2020	2; females; 12 and 16 (dizygotic twins)			Generalized maculopapular rash (2/2)	Mycoplasma (1/2)	No specific therapy (1/2) Oral prednisone (1/2)	Developed alopecia areata (1/2)
Guleria et al. [37] 2020	6; 4 males; 2 females; mean age 10.8			No	NA	Oral prednisolone (2/6)	Concomitant SLE (1/6)
Arslan et al. [46] 2020	1; female; 12			No	No	Oral prednisone	Optic Neuritis
Kim et al. [26] 2020	1; female; 7			No	Parvovirus B19	No specific Therapy	Severe neutropenia
Al Mosawi et al. [12] 2020	11; 6 males; 5 females; mean age 10			Generalized maculopapular rash (4/11)	No	Hidroxychloroquine (7/11) Corticosteroids (1/11) No specific therapy (3/11)	No
Cannon et al. [27] 2020	1; female; 11			No	No	No specific therapy	No
Hua et al. [38] 2021	1; male; 14			No	No	IV methylprednisolone	Appendectomy (mimicked appendicitis)
Yang et al. [14] 2021	13; 9 males, 4 females			Generalized maculopapular rash in 10 patients	ANAs in 4 patients	IVIG + Methylprednisolone: (6/13) Methylprednisolone + Cyclosporine A: (1/13) Cyclosporine A + Dexamethasone + Etoposide: (1/13) Methylprednisolone + Cyclosporine A + Cyclophosphamide: (1/13) Ruxolitinib + Methylprednisolone: (1/13) Methylprednisolone + Dexamethasone: (1/13) Symptomatic treatment: (3/13) Intrathecal Injection + Methotrexate: (1/13) IV Immunoglobulin: (1/13)	Hemophagocytic syndrome (13/13) CNS involvement (8/13) Developed SLE (1/13) Developed SS (2/13)
Danai et al. [13] 2021	1; female; 10			Generalized maculopapular rash and oral ulcers	ANAs	Prednisone Hidroxychloroquine Methotrexate	Concomitant SLE
Öztürk et al. [47] 2021	1; male; 5			No	COVID-19	No specific Therapy	No
Chisholm et al. [28] 2022	14; NA; NA			NA	NA	NA	NA
Takahashi et al. [15] 2023	2; female, 13; male; 11 brothers			Plantar annular erythema and chilblain lupus-like rashes	No (later met the criteria for SLE)	Corticosteroid+ Hydroxychloroquine (1/2) Hydroxychloroquine (1/2)	Developed SLE Developed SS (2/2)
Sevrin et al. [48] 2023	1; male, 12			No	COVID-19 vaccination	Corticosterois	No
Choi Sujin et al. [17] 2023	114; 62 males; 52 females; mean age 12			Oral ulcer (12/114) Generalized rash (11/114)	NA	Corticosteroids (46/114)	No
Zhou et al. [49] 2024	1; male; 4			No	No	Oral prednisone	CASPR2 antibody-associated encephalitis
Lu et al. [50] 2024	1; male; 13			No	No	Corticosterois Rituximab	occlusive retinal vasculitis with near total central retinal artery occlusion
Bao et al. [36] 2024	1; male; 13			No	No	IVIG, Oral prednisone	Optic Neuritis
Harrison et al. [16] 2024	1; female; 16			Generalized rash, sparing the nasolabial folds.	No (later met the criteria for SLE)	Hydroxychloroquine, prednisone, and mycophenolate mofetil	Developed SLE
Camacho-Badilla et al. [20] 2005	1; male; 10	Lymph node excisional biopsy	Granulomas characterized by central necrosis with abundant karyorrhexis, surrounded by histiocytes, lymphocytes and giant multinucleated cells, without neutrophils	No	No	No specific therapy	No
Burns et al. [29] 2020	1; female; 16	FNAC	Necrotizing lymphadenitis	No	EBV, CMV	No specific therapy	No
Sierra et al. [7] 1999	1; male; 14	1. Lymph node excisional biopsy 2. Cutenaous biopsy	1. Histiocytic necrotizing lymphadenitis (see previous) 2. Lymphohistiocytic infiltrates, predominantly perivascular, without any necrosis of the adjacent vascular wall	Generalized maculopapular rash	No	IV Methylprednisolone	Multisystemic involvement Adverse reaction to drugs
Chen et al. [33] 2000	1; female; 13	1. Lymph node excisional biopsy 2. Bone marrow aspirate	1. Histiocytic necrotizing lymphadenitis (see previous) 2.hemophagocytic histiocytes	No	EBV	IVIG IV methylprednisolone oral dexamethasone	hemophagocytic syndrome
Jun-Fen et al. [34] 2007.	1; female; 7	1. Lymph node excisional biopsy 2. Bone marrow aspirate	1. Histiocytic necrotizing lymphadenitis (see previous) 2. Specific Changes In idiopathic thrombocytopenic purpura	No	No	IVIG IV methylprednisolone Oral dexamethasone	Idiopathic thrombocytopenic purpura and Mobitz type II atrioventricular block
Sykes et al. [41] 2016	1; female; 26	1. Lymph node excisional biopsy 2. Bone marrow aspirate	1. Cortical or paracortical necrosis with various histiocytes, plasmacytoid monocytes and a variable number of lymphoid cells with caryorrhectic nuclear fragments and absence of neutrophils 2. hemophagocytic histiocytes	No	No	High-dose corticosteroids, IVIG, and plasmapheresis	Hemophagocytic syndrome
Das et al. [51] 2019	1; male; NA	1. Lymph node excisional biopsy 2. Bone marrow aspirate	1. Cortical or paracortical necrosis with various histiocytes, plasmacytoid monocytes and a variable number of lymphoid cells with caryorrhectic nuclear fragments and absence of neutrophils 2. Reticulo-endothelial activity	Generalized maculopapular rash	Brucella	Rifampicin and gentamicin, doxycycline, trimethoprim-sulfamethoxazole. Oral prednisolone	Doxycycline-induced intracranial hypertension
Lin et al. [19] 2019	40; 20 males; 20 females; mean age 13.9	Lymph node excisional biopsy	14 patients with proliferative type containing karyorrhectic nuclear fragments, eosinophilic apoptotic debris and various histiocytes 23 patients with necrotizing type showing the existence of coagulative necrosis 1 patient with xanthomatous type showing foamy histiocytes. 2 patients with NA histopathology	Generalized maculopapular rash (2/40)	ANAs (1/40) EBV (1/40)	Oral prednisone (5/40) Hidroxychloroquine (5/40) No specific therapy in the rest	No
Abdu et al. [52] 2022	1; female, 10	1. Lymph node excisionalbiopsy 2. Bone marrow biopsy	1. Necrotizing his- tiocytic lymphadenitis 2. Normocellular bone marrow with reactive lymphocytosis and hemophagocytosis	No	Acute otitis media	Corticosterois	No

## 4. Discussion

The currently available literature supports the notion that KFD is not an uncommon condition in childhood. Our case can be considered epidemiologically typical, as it occurs in a female under 30 years of age. While earlier studies suggested a female predominance, more recent research indicates that the incidence of KFD is similar between males and females [1,53]. Based on our review of pediatric cases, a higher proportion of males has been reported. This male predominance in pediatric cases has also been described in other series not included in this review [4].

Cutaneous manifestations associated with KFD have been described as more frequent in children, with an incidence of approximately 10% [4], consistent with the findings in our review. These manifestations typically present as a maculopapular rash, predominantly on the face and the upper trunk. However, a wide range of dermatoses has been documented, including urticarial, morbilliform, rubella-like, lupus-like, or drug-eruption-like rashes; facial erythema; generalized erythema; papules, plaques, and nodules; leukocytoclastic vasculitis; erythema multiforme; papulopustules; and even eyelid edema [54,55,56]. The histopathology of cutaneous lesions in KFD is similarly diverse. Specific findings associated with extranodal KFD include an infiltrate composed of lymphocytes, histiocytes, plasmacytoid monocytes, and nuclear debris, notably in the absence of neutrophils [55,57]. Occasionally, the histology resembles the cutaneous features of SLE, showing interface changes with the vacuolar degeneration of basal cells, necrotic keratynocites and mild perivascular infiltrates of lymphocytes and histiocytes in skin biopsies [54,58]. In any case, the overlap of histologic findings has made it challenging for dermatopathologists to classify a skin biopsy as either extranodal KFD or SLE. In a series of 16 KFD cases with skin biopsy, Kim et al. used direct immunofluorescence negativity and the absence of plasma cells as the primary criteria to rule out SLE [57]. It has been suggested that interface changes observed in skin biopsies may predict an increased risk of developing SLE [59]. This study is based on 10 cases of KFD with cutaneous manifestations and skin biopsy, all of which subsequently developed SLE, with interface dermatitis observed in each biopsy. However, the same authors acknowledge that this statement is not entirely consistent, citing three cases of KFD with interface dermatitis on skin biopsy that did not progress to SLE [59].

Our case presented malar erosive lesions typically affecting the ears, resembling those described in other case reports involving both pediatric and adult patients [60,61]. Histologically, no sufficient evidence of extranodal KFD or SLE was identified; instead, the biopsy revealed a superficial perivascular lymphocytic infiltrate without interface changes. The limited findings in the skin biopsy may be attributed to the location near the eyelid of the sample, chosen to minimize the aesthetic impact of scarring in an area where the erosive component was minimal. Nevertheless, non-specific perivascular dermatitis is a common finding in skin biopsies of patients with KFD [7,62]. Lupus-like rash appears to predict a higher risk of developing SLE. This, combined with the presence of ANAs in the patient’s analysis, prompted the initiation of hydroxychloroquine therapy and the establishment of close follow-up.

Fine needle aspiration of the lymph node does not seem to provide a sufficient sample for diagnosis [2], although it was the technique used in one of the cases in our review [29]. In general, the excisional biopsy of a lymph node is recommended for the diagnosis of KFD. The histopathologic findings of the lymph node biopsy, both in the case presented and in most of the pediatric cases reviewed, are consistently similar: cortical or paracortical necrosis, karyorrhexis, histiocytes (MPO+, CD68+), plasmacytoid dendritic cells (CD123+), and T-cell activation. Notably, there is an absence of neutrophils, eosinophils, and granulomas. Histopathologically, three chronologically related stages of KFD have been described [1,2]. Initially, the proliferative stage is characterized by follicular hyperplasia and lymphohistiocytic infiltrates [1,2]. During the necrotizing stage, karyorrhexis and necrosis artifacts become evident, while the lymph node architecture remains intact [1,2]. Finally, in the xanthomatous stage, foamy histiocytes are observed as the necrotic areas regress [1,2]. Based on these stages, it can be inferred that most pediatric cases presented findings consistent with the proliferative and necrotizing phases, although this distinction was not explicitly made by the authors. Only Lin et al. classified their findings according to these phenotypes, reporting 14 patients with the proliferative type, 23 with the necrotizing type, and 1 with the xanthomatous type [19]. In summary, the histologic findings of lymph node biopsies in pediatric KFD are similar to those observed in adults.

An acute *M. pneumoniae* infection may have triggered the clinical presentation in our patient. Other pediatric and adult cases have also identified this infectious agent as a potential trigger [25,63,64]. Additional agents including *Yersinia*, *Toxoplasma*, EBV, human herpes virus 6 and 8, human T-lymphotropic virus type 1, and parvovirus B19 have been implicated as possible causes, though these associations remain unconfirmed [1,2]. However, the majority of reviewed cases were idiopathic.

KFD is typically a self-limiting condition unless complications arise [1,2]. Antibiotics are frequently administered prior to a definitive diagnosis, as occurred in our case [1,2]. The use of IV immunoglobulin and methylprednisolone appears justified in severe cases, such as those involving CNS complications or hemophagocytic syndrome [9,14,33,34,35,36,37,38,39]. Similarly, hydroxychloroquine has been employed in cases with concomitant SLE or those at high risk of developing it (12,13,15,16,18,1). Nonetheless, clinical trial data remain insufficient to fully validate these therapeutic approaches.

Considering the clinical manifestations, the differential diagnosis of KFD is broad due to the nonspecific nature of its symptoms [1,28]. On one hand, infectious agents could be responsible for lymphadenitis and fever [1,28,57]. On the other hand, immune-mediated diseases, particularly SLE, could explain the systemic and cutaneous findings [1,28,57]. In fact, the concurrent or sequential occurrence of KFD and SLE has been frequently reported. Therefore, ruling out SLE in patients with compatible clinical features or a confirmed diagnosis of KFD is imperative [16,59,64]. Additionally, oncologic and hematologic processes (Classical Hodgkin lymphoma, B-cell or T-cell non-Hodgkin lymphomas, myeloid sarcoma) could account for lymphadenopathy and nonspecific systemic symptoms in pediatric patients [1,22,28,57]. In our case, testing for most infectious agents (CMV, EBV, HIV, HAV…), both in blood and in lymph node samples, returned negative results, except for acute infection with *M. pneumoniae*, which, as previously discussed, may have acted as a triggering factor. The autoimmune findings in our patient, while not sufficient to diagnose concomitant SLE and KFD, do raise the possibility of the future development of SLE. This potential progression justified the initiation of hydroxychloroquine therapy and periodic follow-up. Finally, the possibility of an oncologic or hematologic process, which warranted excisional biopsy of the lymph node, was excluded after histological and immunohistochemical analysis of the specimen.

Considering the histological findings, differentiating KFD from SLE lymphadenopathy is challenging due to overlapping features like the paracortical necrosis and absence of neutrophils [1]. Specific findings in SLE lymphadenopathy include hematoxylin bodies, composed of polysaccharides, immunoglobulins, and nuclear debris [65,66]. The Azzopardi phenomenon, with hematoxylin-stained nuclear material in blood vessels within necrotic areas, can also aid in diagnosis [65,66]. Multiple infectious agents can cause necrotizing lymphadenitis. However, bacterial infections like Yersinia enterocolitica are characterized by abundant neutrophils, while tuberculosis, histoplasmosis, and cat-scratch disease typically show epithelioid histiocyte proliferation with granuloma formation [1,67]. Infectious mononucleosis often displays marked follicular hyperplasia, paracortical expansion, and increased immunoblasts, with scattered Hodgkin- and Reed-Sternberg-like cells [68]. Histologic findings require correlation with serologic and molecular testing for accurate diagnosis [1]. Immunohistochemistry can aid in differentiating KFD from lymphomas [1] but should never replace the hematoxylin and eosin evaluation of an adequate sample, as MPO positivity could be misinterpreted as myeloid sarcoma [69].

It remains necessary to address the differential diagnosis of the cutaneous findings. While maculopapular rash is a common, nonspecific feature in drug- or virus-related eruptions, the predominantly malar distribution and the erosive elements in our patient initially suggested a diagnosis of eczema herpeticum or SLE. However, the absence of cytopathic features and negative molecular studies ruled out eczema herpeticum, and the lack of histologic findings consistent with SLE in H-E staining, IHC and direct immunofluorescence ruled out this diagnosis. Vacuolar degeneration is frequently observed in skin samples from KFD patients, expanding the histopathologic differential diagnosis to papulosquamous disorders, such as pityriasis lichenoides et varioliformis acuta (PLEVA). Nevertheless, PLEVA is typically accompanied by neutrophils and rarely by histiocytes [70]. Given the focus of this manuscript on pediatric cases, Kawasaki disease should also be considered. Although it could justify part of the clinic, histologic findings in this condition include vasculitis, psoriasiform dermatitis, and interface changes resembling erythema multiforme [71]. The absence of neutrophils in the infiltrate excluded conditions that could explain the clinical manifestations, such as acute febrile neutrophilic dermatosis and bacterial soft tissue infections [57].

The present case provides results consistent with an extensive literature review of reported cases. However, some articles lacked detailed descriptions of histological findings, limiting the ability to identify KFD subtypes. The absence of follow-up in certain studies may have led to an underestimation of immune-mediated diseases associated with KFD. Additionally, a lack of specific dermatological examinations in some cases has diminished the value of findings related to cutaneous manifestations.

In conclusion, KFD should be considered in the differential diagnosis of persistent fever and lymphadenopathy in pediatric patients, as cutaneous manifestations are common, often presenting as maculopapular rashes. An excisional biopsy of a lymph node is essential for a definitive diagnosis, though a skin biopsy can provide valuable diagnostic information and help identify patients at risk of developing SLE.

## Figures and Tables

**Figure 1 dermatopathology-12-00007-f001:**
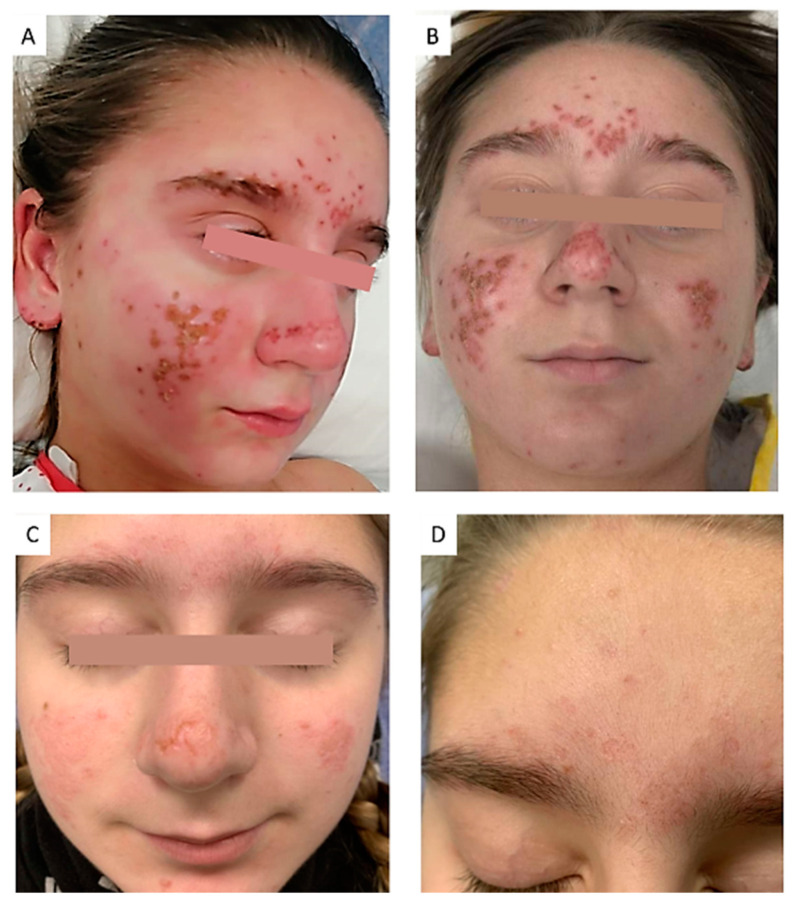
Cutaneous manifestations: (**A**) Onset of the cutaneous eruption, characterized by erosive lesions on erythematoedematous plaques distributed in the centrofacial area and auricular lobes. (**B**) Resolution of the erythematoedematous component, persistence of ulcers and crusts. (**C**) One week after hospital discharge, progressive remission and re-epithelialization of the erosions. (**D**) One month after hospital discharge, showing residual marks of the rash resembling varioliform scars.

**Figure 2 dermatopathology-12-00007-f002:**
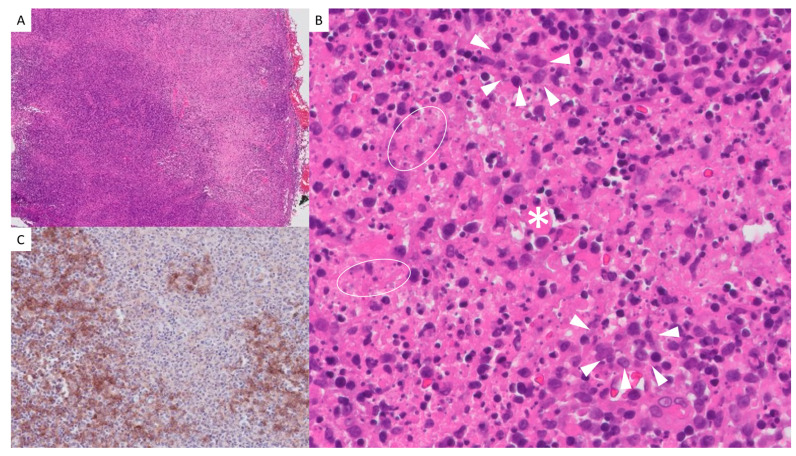
Lymph node biopsy: (**A**) Hematoxylin and eosin (H&E) stain 20×: Normal lymphoid node population in the left half, with areas of paracortical necrosis. Extensive necrosis is observed in the right half. (**B**) H&E 400×: Area of central necrosis lacking polymorphonuclear cells (asterisk), flanked by an infiltrate rich in immunoblasts, plasmacytoid dendritic cells, and histiocytes (arrowheads), without forming organized granulomas. Nuclear debris (white circles) is present within the necrotic area. (**C**) Immunohistochemistry (IHC) 100×: CD123 expression in plasmacytoid dendritic cells surrounding the areas of necrosis.

**Figure 3 dermatopathology-12-00007-f003:**
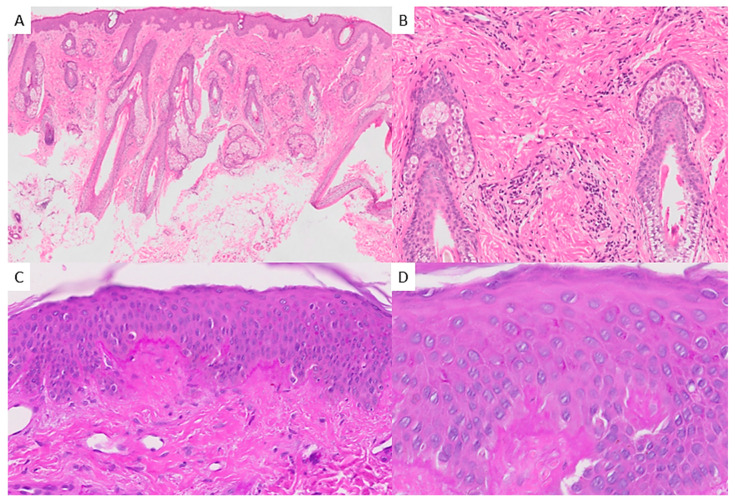
Skin biopsy: (**A**) H&E, 2×: superficial and mild lymphocytic perivascular dermatitis, lack of interface changes, adnexal trophism, mucin deposits or nuclear debris. (**B**) H&E, 10×: lymphocytic infiltrate without peri adnexal involvement and no neutrophils present. (**C**) PAS stain, 20×: focal thickening of the basal layer. (**D**) PAS stain, 40×: mild spongiosis without vacuolar degeneration of basal cells.

**Figure 4 dermatopathology-12-00007-f004:**
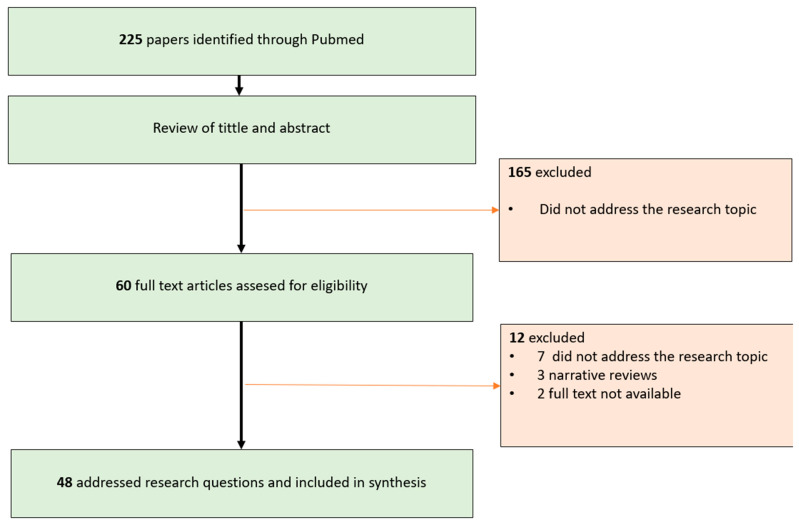
Research strategy and selected articles.

## Data Availability

Data supporting the conclusions of this study are available upon request from the corresponding author.

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
