# Peer review of "Pediatric Kikuchi–Fujimoto Disease: Case Report and Review of Cutaneous and Histopathologic Features in Childhood"

_dermatopathology, 2025, doi:10.3390/dermatopathology12010007_

Round 1

Reviewer 1 Report

Comments and Suggestions for Authors

Major revisions:

The authors report the case
of Pediatric Kikuchi-Fujimoto disease with malar erythema. The article
is of interest however, some changes are needed:

-
Please add the histological images of the cutaneous biopsy
-
Did you perform also in the cutaneous specimen immunohistochemical essays?
-
How do you explain the cutaneous lesions?
-
Did you test the patient for EBV, CMV, and autoimmune diseases?
-
Please add the main clinic and pathologic differential diagnoses

Author Response

Reviewer  1
Major revisions:
The authors report the case of Pediatric Kikuchi-Fujimoto disease with malar erythema. The article is of interest however, some changes are needed:
- Please add the histological images of the cutaneous biopsy 

 Dear Reviewer,

In accordance with your recommendations, we have added a histological figure of the skin biopsy to the manuscript.

 Thank you for your valuable suggestions to enhance the manuscript by recommending additional material.

- Did you perform also in the cutaneous specimen immunohistochemical essays?

 The skin biopsy underwent direct immunofluorescence and immunohistochemical studies. Direct immunofluorescence findings were negative, immunohistochemistry showed no CD123 dendritic cells, these findings were helpful in ruling out SLE, but the lack of plasmacytoid and nuclear debris elements makes it difficult to interpret the results. We therefore believe that the cutaneous findings fall within the category of nonspecific biopsy results that are typical of most cutaneous manifestations of KFD.   The histologic findings of the skin biopsy are detailed further in the Results section.

- How do you explain the cutaneous lesions?

 According to the reviewed series, the most common histopathologic findings in KFD are superficial and deep lymphohistiocytic, predominantly histiocytic infiltrates with nonneutrophilic nuclear debris and interface changes (Kim et al., 2010; Spies et al., 1999). However, these findings do not always appear simultaneously, leading to biopsies where only a nonspecific superficial perivascular lymphohistiocytic infiltrate is observed, rendering these findings nonspecific and limiting the diagnosis of extranodal KFD.  Occasionally, histologic features characteristic of SLE are observed in KFD with asocciated skin lesions. These findings could support a concomitant diagnosis of SLE or may indicate a predisposition to developing SLE in the future.

 In our patient, the findings appear to align more closely with the first group. However, it is possible that prior treatments and the location of the biopsy may have affected the quality of the results. We have included a dedicated section within the discussion to elaborate on the nature of these findings:

“The limited findings in the skin biopsy may be attributed to the location of the sample, chosen to minimize the aesthetic impact of scarring in an area where the erosive component was minimal. Nevertheless, non-specific perivascular dermatitis is a common finding in skin biopsies of patients with KFD (7,61).”

 The low number of fungal spores and their lack of correlation with the other findings led to the interpretation, upon rechecking the biopsy, that they represent contamination by saprophytic spores. Given the irrelevance of this incidental finding, it has been excluded from the text.

- Did you test the patient for EBV, CMV, and autoimmune diseases?

Dear reviewer,

Serologic tests for infectious agents (both bacterial and viral) that could explain the clinical presentation, as well as an autoimmune profile including antinuclear antibodies and serum complement levels, were conducted. The corresponding results are included in the body of the text, specifically within the Results section.

“Serology tests for cytomegalovirus (CMV), Ebstein-Barr virus (EBV), human immunodeficiency virus (HIV), Hepatitis A, B, and C, Toxoplasma, Treponema, Bartonella, Brucella, and Leishmania were all negative. However, IgM antibodies against Mycoplasma pneumoniae were identified, along with positive antinuclear antibodies (ANAs) (speckled pattern,1:160).”- Lines 101-104.

Studies to identify infectious agents were performed on the excisional lymph node biopsy specimen, with the following results:

“Immunohistochemical staining for CMV was negative. The in-situ hybridization study for the detection of EBV RNA was also negative. PAS, Giemsa, Grocott, and Warthin-Starry stains, along with immunohistochemical staining for spirochetes, did not detect microorganisms in the examined sections.” Lines 132-134.

Additionally, a section of the Discussion has been dedicated to addressing the relevance of these findings and the potential role of Mycoplasma pneumoniae as a triggering agent.

“An acute M. pneumoniae infection may have triggered the clinical presentation in our patient. Other pediatric and adult cases have also identified this infectious agent as a potential trigger(25,62,63). Additional agents including Yersinia, Toxoplasma, EBV, human herpes virus 6 and 8, human T-lymphotropic virus type 1, and parvovirus B19  have been implicated as possible causes, though these associations remain unconfirmed (1,2).  However, the majority of reviewed cases were idiopathic.” Lines 286-290.

Thank you for improving the quality of the manuscript.

- Please add the main clinic and pathologic differential diagnoses

 In accordance with your recommendations, we have added an additional section to the Discussion where the clinical and histologic differential diagnoses are thoroughly addressed. To support this, we conducted an additional literature review. The following is the text concerning the differential diagnosis, which will be available in the Discussion (lines 298-340).

“Considering the clinical manifestations, the differential diagnosis of KFD is broad due to the nonspecific nature of its symptoms(1,28). On one hand, infectious agents could be responsible for lymphadenitis and fever(1,28,64). On the other hand, immune-mediated diseases, particularly SLE, could explain the systemic and cutaneous findings(1,28,64). In fact, the concurrent or sequential occurrence of KFD and SLE has been frequently reported. Therefore, ruling out SLE in patients with compatible clinical features or a confirmed diagnosis of KFD is imperative(16,58,63). Additionally, oncologic and hematologic processes (Classical Hodgkin lymphoma, B-cell or T-cell non-Hodgkin lymphomas, myeloid sarcoma) could account for lymphadenopathy and nonspecific systemic symptoms in pediatric patients(1,22,28,64). In our case, testing for most infectious agents (CMV, EBV, HIV, HAV…), both in blood and in lymph node samples, returned negative results, except for acute infection with M. pneumoniae, which, as previously discussed, may have acted as a triggering factor. The autoimmune findings in our patient, while not sufficient to diagnose concomitant SLE and KFD, do raise the possibility of future development of SLE. This potential progression justified the initiation of hydroxychloroquine therapy and periodic follow-up. Finally, the possibility of an oncologic or hematologic process, which warranted excisional biopsy of the lymph node, was excluded after histological and immunohistochemical analysis of the specimen.

Considering the histological findings, differentiating KFD from SLE lymphadenopathy is challenging due to overlapping features like paracortical necrosis and absence of neutrophils(1). Specific findings in SLE lymphadenopathy include hematoxylin bodies, composed of polysaccharides, immunoglobulins, and nuclear debris(65,66). The Azzopardi phenomenon, with hematoxylin-stained nuclear material in blood vessels within necrotic areas, can also aid in diagnosis(65,66). Multiple infectious agents can cause necrotizing lymphadenitis. However, bacterial infections like Yersinia enterocolitica are characterized by abundant neutrophils, while tuberculosis, histoplasmosis, and cat-scratch disease typically show epithelioid histiocyte proliferation with granuloma formation (1,67). Infectious mononucleosis often displays marked follicular hyperplasia, paracortical expansion, and increased immunoblasts, with scattered Hodgkin- and Reed-Sternberg-like cells(68). Histologic findings require correlation with serologic and molecular testing for accurate diagnosis(1).   Immunohistochemistry can aid in differentiating KFD from lymphomas(1) but should never replace hematoxylin and eosin evaluation of an adequate sample, as MPO positivity could be misinterpreted as myeloid sarcoma(69).

It remains necessary to address the differential diagnosis of the cutaneous findings. While maculopapular rash is a common nonspecific feature in drug- or virus-related eruptions, the predominantly malar distribution and the erosive elements in our patient initially suggested a diagnosis of eczema herpeticum or SLE. However, the absence of cytopathic features, and negative molecular studies ruled out eczema herpeticum, and lack of histologic findings consistent with SLE in H-E staining, IHC and direct immunofluorescence ruled out this diagnosis.  Vacuolar degeneration is frequently observed in skin samples from KFD patients, expanding the histopathologic differential diagnosis to papulosquamous disorders, such as pityriasis lichenoides et varioliformis acuta (PLEVA). Nevertheless, PLEVA is typically accompanied by neutrophils and rarely by histiocytes (70). Given the focus of this manuscript on pediatric cases, Kawasaki disease should also be considered. Although it could justify part of the clinic, histologic findings in this condition include vasculitis, psoriasiform dermatitis, and interface changes resembling erythema multiforme (71). The absence of neutrophils in the infiltrate excluded conditions that could explain the clinical manifestations, such as acute febrile neutrophilic dermatosis and bacterial soft tissue infections (64).”

Thank you for pointing this out.

Reviewer 2 Report

Comments and Suggestions for Authors

This is a well-written case report titled "Pediatric Kikuchi-Fujimoto Disease with Malar erythema: a case report and review of Cutaneous and histopathologic presentations in childhood." The authors present a case of an 11-year-old girl who presented with fevers and lymphadenopathy and subsequently developed erosive lesions on the face. Based on the histopathologic features seen in the lymph node excision, the case was diagnosed with Kikuchi-Fujimoto. The authors review the literature and provide a detailed summary in a table format.

The only comment for the authors is to change the term "ciliary" to the proper term in English.

Author Response

Reviewer  2

This is a well-written case report titled "Pediatric Kikuchi-Fujimoto Disease with Malar erythema: a case report and review of Cutaneous and histopathologic presentations in childhood." The authors present a case of an 11-year-old girl who presented with fevers and lymphadenopathy and subsequently developed erosive lesions on the face. Based on the histopathologic features seen in the lymph node excision, the case was diagnosed with Kikuchi-Fujimoto. The authors review the literature and provide a detailed summary in a table format.

The only comment for the authors is to change the term "ciliary" to the proper term in English.

 Dear Reviewer,

In accordance with your recommendations, the term "ciliary" has been removed and replaced with a more appropriate term. We sincerely appreciate your thoughtful comments and the positive assessment of the manuscript. We are delighted to know that you found the manuscript useful.

Reviewer 3 Report

Comments and Suggestions for Authors

I have reviewed the manuscript "Pediatric Kikuchi-Fujimoto Disease with Malar Erythema: A Case Report and Review of Cutaneous and Histopathologic Presentations in Childhood." It is engaging and well-written.

1. As the skin biopsy was performed, one low-power and one high-power image of the histopathology of the skin biopsy will be helpful for the readers.

2. The skin histopathology in Pediatric KFD appears non-specific, with some cases showing LE-like features. As this is a dermatopathology journal, more emphasis must be placed on that message (e.g., a dedicated paragraph in the discussion, mention in the abstract, etc.). What will be the take-home message for dermatopathologists reading this article?

Thanks.

Author Response

Reviewer  3

I have reviewed the manuscript "Pediatric Kikuchi-Fujimoto Disease with Malar Erythema: A Case Report and Review of Cutaneous and Histopathologic Presentations in Childhood." It is engaging and well-written.

  1. As the skin biopsy was performed, one low-power and one high-power image of the histopathology of the skin biopsy will be helpful for the readers.

Dear Reviewer,

In accordance with your recommendations, we have added a histological figure of the skin biopsy to the manuscript.

 Thank you for your valuable suggestions to enhance the manuscript by recommending additional material.

  1. The skin histopathology in Pediatric KFD appears non-specific, with some cases showing LE-like features. As this is a dermatopathology journal, more emphasis must be placed on that message (e.g., a dedicated paragraph in the discussion, mention in the abstract, etc.). What will be the take-home message for dermatopathologists reading this article? Thanks.

 Thank you for your recommendation. To enhance the usefulness of the manuscript, we have added a comment on the value of skin biopsy in the abstract (Lines 51-52). Additionally, we have expanded the relevant section in the discussion (Lines 242-269). Our goal is not only to encourage dermatologists to consider this entity in pediatric cases, but also to improve the interpretation of histologic results from skin biopsies. We emphasize the diagnostic and prognostic value of these results, outlining key findings that dermatopathologists should look for. Below is the additional text from the discussion for your review:

“The histopathology of cutaneous lesions in KFD is similarly diverse. Specific findings associated with extranodal KFD include an infiltrate composed of lymphocytes, histiocytes, plasmacytoid monocytes, and nuclear debris, notably in the absence of neutrophils (55,57). Occasionally, the histology resembles the cutaneous features of SLE, showing interface changes with vacuolar degeneration of basal cells, necrotic keratynocites and mild perivascular infiltrates of lymphocytes and histiocytes in skin biopsies (54,58). In any case, the overlap of histologic findings has made it challenging for dermatopathologists to classify a skin biopsy as either extranodal KFD or SLE. In a series of 16 KFD cases with skin biopsy, Kim et al. used direct immunofluorescence negativity and the absence of plasma cells as the primary criteria to rule out SLE(57).  It has been suggested that interface changes observed in skin biopsies may predict an increased risk of developing SLE(59).  This study is based on 10 cases of KFD with cutaneous manifestations and skin biopsy, all of which subsequently developed SLE, with interface dermatitis observed in each biopsy. However, the same authors acknowledge that this statement is not entirely consistent, citing three cases of KFD with interface dermatitis on skin biopsy that did not progress to SLE(59).”

Additionally, we have incorporated an extra paragraph including other differential diagnoses to be taken into account in the histopathologic evaluation of skin manifestations (lines 326-340):

“It remains necessary to address the differential diagnosis of the cutaneous findings. While maculopapular rash is a common nonspecific feature in drug- or virus-related eruptions, the predominantly malar distribution and the erosive elements in our patient initially suggested a diagnosis of eczema herpeticum or SLE. However, the absence of cytopathic features, and negative molecular studies ruled out eczema herpeticum, and lack of histologic findings consistent with SLE in H-E staining, IHC and direct immunofluorescence ruled out this diagnosis. Vacuolar degeneration is frequently observed in skin samples from KFD patients, expanding the histopathologic differential diagnosis to papulosquamous disorders, such as pityriasis lichenoides et varioliformis acuta (PLEVA). Nevertheless, PLEVA is typically accompanied by neutrophils and rarely by histiocytes(70). Given the focus of this manuscript on pediatric cases, Kawasaki disease should also be considered. Although it could justify part of the clinic, histologic findings in this condition include vasculitis, psoriasiform dermatitis, and interface changes resembling erythema multiforme(71). The absence of neutrophils in the infiltrate excluded conditions that could explain the clinical manifestations, such as acute febrile neutrophilic dermatosis and bacterial soft tissue infections(57).”

We would like to sincerely thank you for your recommendations to improve the quality of the article.

Reviewer 4 Report

Comments and Suggestions for Authors

The article is interesting but the differential diagnoses must to be better defined as Sweet S, cat scrath disease, hydroa vacciniforme, photodermatoses.... The skin biopsy is important only to rule out or confirm a connective tissue disease?How long do you use a hydrossicloroquine in case of a sospected LES?

Author Response

Reviewer  4

The article is interesting but the differential diagnoses must to be better defined as Sweet S, cat scrath disease, hydroa vacciniforme, photodermatoses....

Estimated reviewer,

In accordance with your recommendations, we have added an additional section to the Discussion where the clinical and histologic differential diagnoses are thoroughly addressed. To support this, we conducted an additional literature review. The text concerning the differential diagnosis will be available in the Discussion (lines 298-340). The following is the section on differential diagnosis with other dermatoses (lines 326-340):

“It remains necessary to address the differential diagnosis of the cutaneous findings. While maculopapular rash is a common nonspecific feature in drug- or virus-related eruptions, the predominantly malar distribution and the erosive elements in our patient initially suggested a diagnosis of eczema herpeticum or SLE. However, the absence of cytopathic features, and negative molecular studies ruled out eczema herpeticum, and lack of histologic findings consistent with SLE in H-E staining, IHC and direct immunofluorescence ruled out this diagnosis. Vacuolar degeneration is frequently observed in skin samples from KFD patients, expanding the histopathologic differential diagnosis to papulosquamous disorders, such as pityriasis lichenoides et varioliformis acuta (PLEVA). Nevertheless, PLEVA is typically accompanied by neutrophils and rarely by histiocytes(70). Given the focus of this manuscript on pediatric cases, Kawasaki disease should also be considered. Although it could justify part of the clinic, histologic findings in this condition include vasculitis, psoriasiform dermatitis, and interface changes resembling erythema multiforme(71). The absence of neutrophils in the infiltrate excluded conditions that could explain the clinical manifestations, such as acute febrile neutrophilic dermatosis and bacterial soft tissue infections(57).”

Thank you for help us improve the quality of the manuscript.

The skin biopsy is important only to rule out or confirm a connective tissue disease?

The usefulness of skin biopsy in KFD lies not only in ruling out other diagnoses, but also in supporting the diagnosis of KFD itself and providing valuable prognostic information. On one hand, identifying a perivascular lymphohistiocytic inflammatory infiltrate with nuclear debris and interface changes in the absence of neutrophils can aid in diagnosing KFD in the appropriate clinical context. On the other hand, the presence of interface changes could indicate a higher likelihood of developing SLE in the future, even if the skin biopsy does not reveal other typical findings of SLE, such as plasma cells, and direct immunofluorescence is negative.

How long do you use a hydrossicloroquine in case of a sospected LES?

 The duration of treatment with hydroxychloroquine in pediatric KFD is not standardized, as it is a rare condition and its effectiveness is primarily based on real-world clinical reports. Therefore, we cannot establish a fixed treatment duration; the decision to continue therapy will depend largely on the clinical presentation. We believe that the absence of clinical signs during follow-up visits will allow for dose reduction and, eventually, discontinuation of treatment within the next 6-12 months. In any case, the patient will continue regular check-ups to ensure that she does not develop SLE.

Round 2

Reviewer 1 Report

Comments and Suggestions for Authors

The authors improved the manuscript and the article can be accepted.

Thank you.